# Cost-effectiveness of point-of-care C-Reactive Protein test compared to current clinical practice as an intervention to improve antibiotic prescription in malaria-negative patients in Afghanistan

**Simon Dickinson**[1,2]*, **Huey Yi Chong**[2], **Toby Leslie**[1], **Mark Rowland**[3], **Kristian Schultz Hansen**[4], **Dwayne Boyers**[2]

**1** Mott MacDonald Ltd, London, United Kingdom, **2** Health Economics Research Unit, University of Aberdeen, Aberdeen, United Kingdom, **3** Department of Disease Control, London School of Hygiene and Tropical Medicine, London, United Kingdom, **4** Department of Public Health, University of Copenhagen, Copenhagen, Denmark

* simon.dickinson85@gmail.com

## Abstract

### Background

Antimicrobial resistance (AMR) is a global health problem requiring a reduction in inappropriate antibiotic prescribing. Point-of-Care C-Reactive Protein (POCCRP) tests could distinguish between bacterial and non-bacterial causes of fever in malaria-negative patients and thus reduce inappropriate antibiotic prescribing. However, the cost-effectiveness of POCCRP testing is unclear in low-income settings.

### Methods

A decision tree model was used to estimate cost-effectiveness of POCCRP versus current clinical practice at primary healthcare facilities in Afghanistan. Data were analysed from healthcare delivery and societal perspectives. Costs were reported in 2019 USD. Effectiveness was measured as correctly treated febrile malaria-negative patient. Cost, effectiveness and diagnostic accuracy parameters were obtained from primary data from a cost-effectiveness study on malaria rapid diagnostic tests in Afghanistan and supplemented with POCCRP-specific data sourced from the literature. Incremental cost-effectiveness ratios (ICERs) reported the additional cost per additional correctly treated febrile malaria-negative patient over a 28-day time horizon. Univariate and probabilistic sensitivity analyses examined the impact of uncertainty of parameter inputs. Scenario analysis included economic cost of AMR per antibiotic prescription.

### Results

The model predicts that POCCRP intervention would result in 137 fewer antibiotic prescriptions (6%) with a 12% reduction (279 prescriptions) in inappropriate prescriptions compared

**Data Availability Statement:** All relevant data are within the paper and its Supporting Information files.

**Funding:** Publication fees for the article are paid by Mott MacDonald Ltd. The authors received no additional funding for this article. Two of the authors (SD and TL) are affiliated to a commercial organisation (Mott MacDonald Ltd). We confirm there is no commercial affiliation between Mott MacDonald and this article. Aside from publication fees, no funding was provided for the production of the article. Three of the authors (SD, HYU and DB) are affiliated to the Health Economics Research Unit at the University of Aberdeen, which is funded by the Chief Scientist's office of the Scottish government health directorates. The primary data collection (2009–2012) for the malaria RDT IRT (Ref: 30) and CEA (Ref: 41) in Afghanistan was funded by the Bill and Melinda Gates Foundation through a grant to the London School of Hygiene and Tropical Medicine. Data on aetiological agents of disease (Ref: 14) was collected with funding from the UK Defence Science and Technology Laboratory. These funding organisations did not play a role in the study design, data collection and analysis, decision to publish, or preparation of the manuscript and only provided financial support in the form of authors' salaries and/or research materials.

**Competing interests:** The commercial affiliation of two authors (SD and TL) does not alter our adherence to PLOS ONE policies on sharing data and materials

to current clinical practice. ICERs were $14.33 (healthcare delivery), $11.40 (societal), and $9.78 (scenario analysis) per additional correctly treated case.

## Conclusions

POCCRP tests could improve antibiotic prescribing among malaria-negative patients in Afghanistan. Cost-effectiveness depends in part on willingness to pay for reductions in inappropriate antibiotic prescribing that will only have modest impact on immediate clinical outcomes but may have long-term benefits in reducing overuse of antibiotics. A reduction in the overuse of antibiotics is needed and POCCRP tests may add to other interventions in achieving this aim. Assessment of willingness to pay among policy makers and donors and undertaking operational trials will help determine cost-effectiveness and assist decision making.

## Introduction

Antimicrobial resistance (AMR) is recognised as a global health security threat [1–6]. There are concerns that the world may return to a pre-antimicrobial era where infectious diseases could become untreatable [5] and many common medical procedures become much riskier to undertake due to risk of incurable infection [1]. It is estimated that by 2050, ten million people per year will die as a result of AMR [6] and annual global GDP could reduce by 1.1% relative to a scenario with no AMR effect, with low and middle income countries (LMICs) expected to be worst affected due to higher prevalence of infectious diseases and weaker health systems [7]. Overuse and misuse of antimicrobial medicines are considered the lead driver of AMR so efforts to reduce the over prescription of antibiotics is essential [8, 9].

In recent years, there has been improvement in diagnosis of malaria following the introduction of malaria Rapid Diagnostic Tests (RDTs) [10]. However, evidence suggests the improvement in malaria diagnosis could be a driver for increased and unnecessary use of antibiotics [11, 12]. Antibiotic prescription is often the default treatment decision for malaria-negative febrile patients because symptoms for bacterial infection are difficult to distinguish from other infections (e.g. viral, fungal, parasitic) [13]. In the majority of cases, the prescription is unnecessary as evidence suggests only a small proportion (10–15%) would have a bacterial infection [14, 15].

In suspected bacterial infections, culture remains the gold standard for detection and treatment [16]. This does not routinely happen in most malaria endemic countries because of the absence of culturing facilities at the periphery of health systems where most malaria patients are treated. Where testing is available, the time taken for samples to be processed is incompatible with the pressure on the clinician to treat or the unaffordability of laboratory services to patients [17]. Point-of-Care C-Reactive Protein (POCCRP) tests are a potential solution to ascertain whether a patient has a bacterial infection and if an antibiotic is appropriate treatment [18–24]. POCCRP could prove useful in LMICs for reducing unnecessary selection of antibiotic resistance where laboratory services are limited and antibiotics are available over the counter without prescription.

C-Reactive Protein (CRP) is a non-specific biomarker of inflammation, which gives an indication of likely bacterial infection [21]. POCCRP tests are rapid tests that detect CRP concentration in the blood at a given concentration, providing a result within 15 minutes [22].

Research to date on POCCRP has focused on high income countries and respiratory tract infections (RTIs) [19, 20, 23, 25, 26]. There has been limited research in LMICs on febrile illness of uncertain cause [18, 22, 27]. POCCRP tests have been shown to be effective in reducing unnecessary use of antibiotics in patients with RTIs [19, 25, 28, 29]. There is insufficient data to indicate that POCCRP tests would reduce unnecessary antibiotic prescriptions in LMICs in febrile patients who turn out not to need them [18, 21].

Economic analysis evidence indicates that POCCRP testing could be cost effective [20, 23, 25, 27]. In Norway and Sweden, it was estimated POCCRP testing had a cost per quality-adjusted life year (QALY) gained of €9391, with a 70% probability of being cost-effective at a willingness-to-pay (WTP) threshold of €30,000 per QALY gained [23]. In a LMIC setting, one study in Vietnam found that introduction of POCCRP was unlikely to be cost-saving due to high non-adherence rate to negative test results [27]. To our knowledge, there is no evidence that compares the costs and outcomes, in an economic evaluation framework, of POCCRP in a LMIC setting. Afghanistan is an example of a country of extreme poverty, endemic for vivax and falciparum malaria, with a shortage of laboratory facilities, but is also a place where the majority of febrile illness is not malarial but more likely to be bacterial or viral in origin [11, 14, 30]. It is in this type of epidemiological setting with moderate to low malaria risk where a POCCRP test might prove of great service for diagnosis and treatment. The aim of this paper was to determine the cost-effectiveness of POCCRP test compared to current clinical practice for malaria-negative febrile patients in Afghanistan, using a pre-existing primary dataset on costs of introducing malaria RDTs and secondary data sources on POCCRP tests.

## Materials and methods

### Study design

A cost-effectiveness analysis (CEA) framed in the evidence-based medicine PICO framework [31]: the Population was febrile malaria negative cases in 22 primary health care facilities in two provinces of Afghanistan [30]; the Intervention was a POCCRP test following a negative malaria diagnosis; the Comparison group was "current clinical practice" after a febrile patient is diagnosed as malaria-negative, where diagnosis is based on patient symptoms using treatment guideline [32]; the Outcome was febrile malaria-negative patient correctly treated. A correctly treated case was defined as: 1) bacterial infection treated by an antibiotic; 2) other cause of fever not treated by an antibiotic.

A decision tree model, developed in Microsoft® Excel® Version 2002, was used to conduct a CEA from two perspectives: public healthcare and societal (incorporating patient and carer costs in addition to health sector costs). A further scenario analysis was conducted to include estimated economic cost of AMR per prescription of antibiotics. The model time horizon was 28-days post initial consultation, deemed sufficient to capture the duration of the visit to the healthcare facility and subsequent costs for patients and carers. Beyond 28 days, it was assumed that all costs and outcomes were independent of the testing pathway chosen. Discounting of costs and outcomes was not required given the short time span.

### Population

The population for the economic model cohort was taken from primary data collected during an individual randomised trial (IRT) comparing effectiveness of malaria diagnostic interventions in primary health clinics in Afghanistan [30]. The POCCRP CEA included a population of 4,391 patients (mean age, 17.9 years). Data for the IRT were collected between 23 September 2009 and 22 September 2010 [30]. Inclusion criteria from the dataset was a patient diagnosed as malaria-negative after being randomised to either the malaria RDT or clinical microscopy

arms of the IRT. The IRT had a third arm which diagnosed by clinical symptoms but this arm was not considered in the model given the extremely high false positive rates for suspected malaria patients diagnosed by clinical signs and symptoms alone [11].

## POCCRP test intervention and accuracy

It is important to note that the POCCRP test detects CRP concentration in the blood, not specifically a bacterial infection. A given concentration level of CRP can provide an indication of bacterial infection but not all bacterial infection patients will exhibit increased CRP levels; similarly, patients with other causes of infection will not always exhibit lower levels of CRP [21]. Thus, the CRP test can be seen as a clinical decision making tool, rather than a diagnostic test for bacterial infection. Accordingly, the model accounts for: 1) the probability that a 'bacterial infection' patient would have CRP concentration of 10 mg/L, and the probability of an 'other cause' patient having a CRP concentration of less than 10 mg/L; and 2) the sensitivity and specificity of the POCCRP test to detect CRP concentration at 10 mg/L.

DTS233 POCCRP test was chosen for the study because of its 10mg/L threshold for a positive result and its simple visual interpretation [22]. 10 mg/L threshold was chosen due to the remote setting of the study. By increasing the CRP threshold, it would decrease the number of false positives but also decrease true positives. As there would be limited possibility for patients to obtain emergency care it was advisable to accept a higher rate of false positives [22]. DTS233's visual interpretation for positive, negative or invalid results is very simple compared to other POCCRP tests (see Fig 1) [22].

## Reference standards

Tests used for clinical decision making must be compared to a reference standard to determine sensitivity and specificity [33]. In this study, Sensitivity and specificity of the POCCRP test to

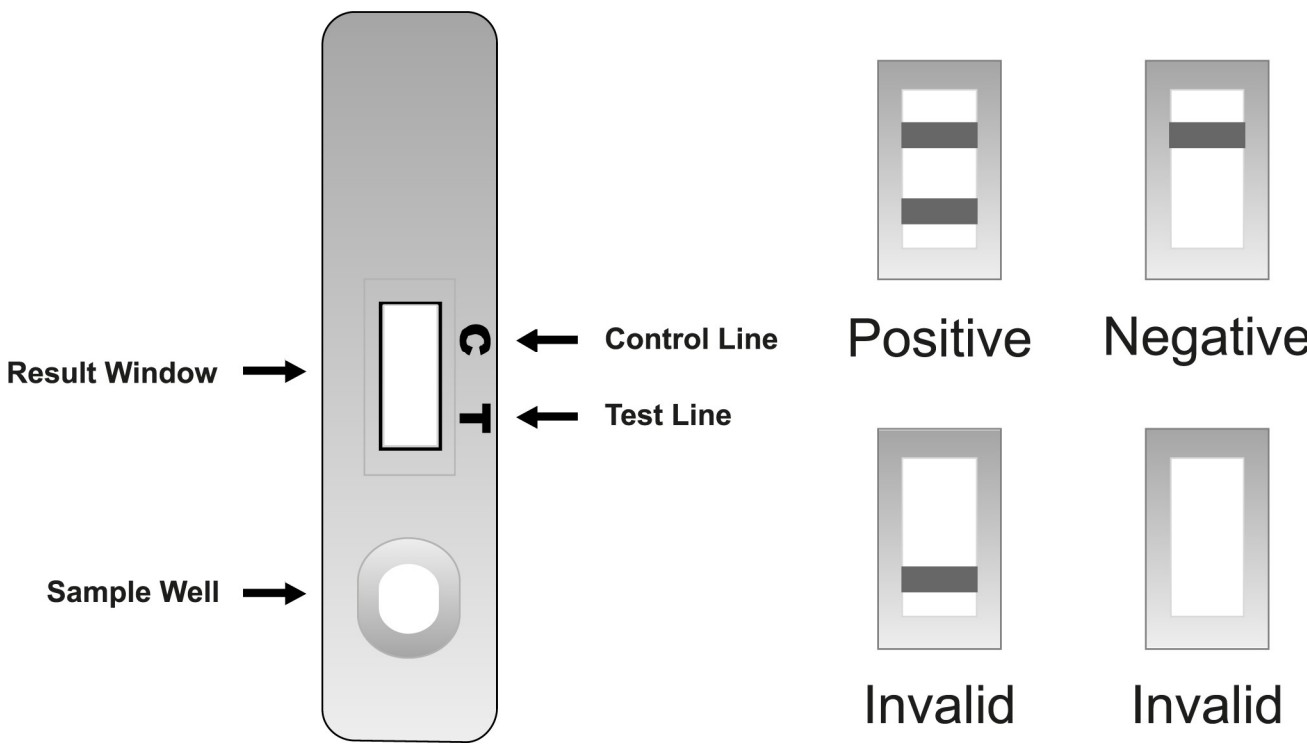

**Fig 1. DTS233 visual interpretation [22].**

detect CRP is cited from a study in rural Laos, where a NycoCard Reader II (immunochemical assay for the quantitative determination of CRP) was used as the reference standard to provide information as to how sensitive and specific the POCCRP test is to detect CRP at 10mg/L does not factor in sensitivity and specific of the NycoCard Reader [22]. At greater than or equal to 10 mg/L, NycoCard Reader II is estimated to have a sensitivity of 94.4% and specificity of 83.3% [34]. Sensitivity and specificity of the NycoCard Reader is not accounted for in the study model.

Laboratory microbiological diagnosis was the reference standard used to assess the probability of a bacterial infection patient having CRP concentration of 10 mg/L or greater, and probability of a non-bacterial infection patient having a CRP concentration of less than 10 mg/L [21].

## Outcome measure

The outcome measured was the number of febrile malaria-negative patients correctly treated. A correctly treated case was defined as: 1) bacterial infection treated by an antibiotic; 2) other cause of fever not treated by an antibiotic.

## Model structure

A decision tree model was developed to compare two approaches to treating malaria-negative febrile patients in Afghanistan. This model structure was chosen as the febrile condition has a short, fixed timeline with no transition between disease states. Fig 2 shows the decision tree with treatment pathways for the two study arms: Arm 1 –current clinical practice where treatment was based on symptoms; and Arm 2 –POCCRP intervention where treatment was based on the result of the POCCRP test.

The model started with a febrile patient who has been diagnosed as malaria-negative. The cause of the fever could either be a bacterial infection or other cause. In arm 1, the patient received a diagnosis and treatment based on symptoms. Whilst in arm 2, the patient received a POCCRP test to guide the diagnosis. Treatment was given according to the test result, where a CRP concentration of greater than or equal to 10mg/L indicated a bacterial infection, and a CRP concentration of less than 10 mg/L indicated other cause. Arm 2 has a stage to factor in probability that a patient will have a CRP concentration of greater than or equal to 10 mg/L CRP, or less than 10mg/L CRP; this is important as not all bacterial infections will have a CRP concentration of 10mg/L or above and not all non-bacterial infections will show a CRP concentration below 10 mg/L. For a bacterial infection diagnosis, an antibiotic prescription is recommended; for other cause of fever diagnosis, no antibiotic was prescribed.

## Model parameters and sources

**Probability parameters.** The prevalence (proportion) of malaria negative patients with a bacterial infection at a particular point in time, was taken from an unpublished study on causes of undifferentiated fever in Afghanistan [14]. This source was chosen as there is a very limited evidence base in non-malaria febrile patients across many countries [15, 37–40]. The probability of being prescribed antibiotic under current clinical practice was derived from the primary dataset for the study on introduction of malaria RDTs in Afghanistan by looking at the proportion of malaria-negative patients that received antibiotics as treatment [30]. Data on CRP cut off points (lower than or greater than CRP 10mg/L) are taken from a study in South East Asia assessing the predictive value of CRP as an indicator for bacterial or non-bacterial infection [21]. Data for sensitivity of POCCRP test for detection of CRP concentration at 10 mg/L and

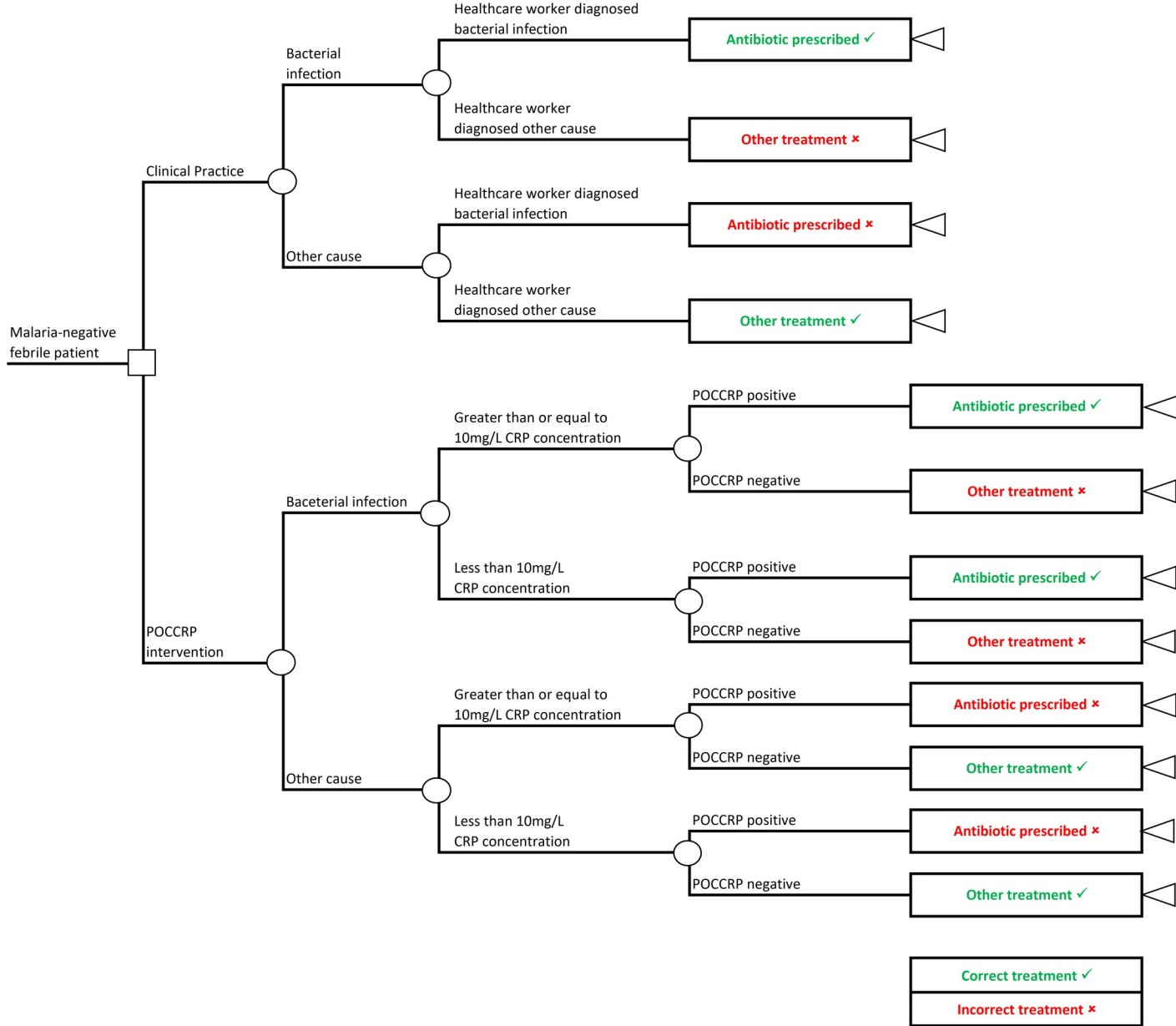

**Fig 2. Decision tree.** The square decision node indicates the decision point between the two arms; circular chance nodes show points where two alternative events for a patients are possible [35]. Branches emanating from each chance node indicate possible events and the sum of their probabilities must equal 1; and triangular terminal nodes are at the ends of pathways and represent the outcome of interest [36]. Recommended treatments are green and incorrect treatments are red. Pathways are mutually exclusive sequences of events [35].

specificity of POCCRP test for non-detection of CRP concentration of less than 10 mg/L is taken from a study on accuracy of commercially available CRP tests [22].

**Cost parameters.** The cost of a POCCRP test kit is taken from an estimate per unit cost, allowing for import tariffs, shipment and other expenses, in a cost comparison study of introduction of POCCRP in the management of acute respiratory infections in the Vietnamese primary healthcare setting [27]. Cost data for training to administer and interpret tests, outpatient services delivery cost, dispensary costs, patient out of pocket expenditure, patient and carer opportunity costs, and cost of a course of antibiotics were all taken from a primary

data set from the cost-effectiveness study of the introduction of malaria RDTs in Afghanistan [41]. We have chosen this study to populate model cost parameters where possible because it represents a sample of patients with fever that has been diagnosed as malaria-negative and is the best available data in the Afghan setting. Economic cost of AMR to society per full course of antibiotics is taken from a study in Thailand on the economic cost of antimicrobial resistance per antibiotic consumed [42]. More detail on model parameters and sources can be found in the table in S2 Appendix: model parameters and sources.

Primary data for the malaria RDT IRT [30] and CEA [41] in Afghanistan was provided by respective lead authors (TL and KSH). Patient data was anonymised. Ethical approval for the data collections has been granted by London School of Hygiene and Tropical Medicine and The Afghan Institutional Review Board. The approvals allow data sharing and use of data for other purposes. The malaria RDT IRT trial and associated CEA was registered at Clinincal-Trials.gov (NCT00935688).

Table 1 shows inputs to build up the decision tree model and distributions assigned for the probabilistic sensitivity analysis (PSA).

Where cost data is taken from a sample set, the standard deviation (SD) and sample size are used to derive 95% confidence intervals. Where cost data is based on a single deterministic value, SD is assumed to be 20% of deterministic value in the absence of a sample from which to derive SD.

Costs were converted to 2019 USD. AFG prices are adjusted accounting for average inflation over 10 years [43] and converted to USD at the exchange rate of 27 August 2019 [44]. The economic cost of AMR data is based an estimate from Thailand [42]. This is adjusted for Afghanistan using purchasing power parity (PPP) rates from April 2019 [45].

## Analysis

**Deterministic analysis.** Using the decision tree model, the expected cost and outcome between the two study arms were estimated. The incremental cost-effectiveness ratio (ICER) was expressed as incremental cost per correctly treated patient, calculated by dividing the incremental cost per patient by the absolute incremental number of correctly treated patients.

$$\text{ICER} = \frac{(C_1 - C_0)}{(E_1 - E_0)}$$

$C_1$ = POCCRP intervention cost
$C_0$ = Current clinical practice cost
$E_1$ = Number of correctly treated patients (POCCRP)
$E_0$ = Number of correctly treated patients (current clinical practice)

**Univariate sensitivity analysis.** Univariate sensitivity analysis was undertaken on the scenario analysis (societal including economic cost of AMR) to assess the impact of each parameter on the ICER and to identify key drivers of the model. One variable was adjusted while all other variables remained constant. Lower and upper bounds were set for each parameter based on upper and lower 95% confidence intervals (CIs). If 95% CIs were not available, parameters were varied by +/- 5%. The range in ICER for each variable between upper and lower bounds was presented in a tornado diagram.

**Probabilistic sensitivity analysis.** A probabilistic sensitivity analysis (PSA) was undertaken to characterise parameter uncertainty and test the robustness of the ICER of the model results to variation in data inputs. Monte Carlo simulation was undertaken by running 1,000 simulations, randomly selecting values from each parameter's distribution simultaneously for each simulation. As all chance nodes only have two branches (binary), beta distributions were

**Table 1. Model inputs.**

| Parameter | Parameter Description | Population (patients) | | | | |
|---|---|---|---|---|---|---|
| Pop | Population | 4391 | | | | |
| **Cost Parameter** | **Parameter Description** | **Mean ($)*** | **SD ($)** | **95% CI** | **Distribution (α, β) +** | **Source** |
| cTEST | Cost of the POCCRP test kit | 1.00 | 0.20 | 0.61–1.39 | Gamma (25, 0.04) | [27] |
| cTRAIN | Training to administer and interpret test | 0.03 | 0.01 | 0.02–0.04 | Gamma (25, 0.001) | [41] |
| cOPD | Outpatient services delivery cost | 0.28 | 0.06 | 0.25–0.31 | Gamma (376.83, 0.001) | [41] |
| cDISP | Dispensary cost at healthcare facility | 1.33 | 0.10 | 1.29–1.38 | Gamma (2.31, 0.58) | [41] |
| cOPEapp | Patient out of pocket expenditure—appropriate treatment | 3.82 | 12.32 | 2.95–4.65 | Gamma (51.64, 33.61) | [41] |
| cPLTapp | Opportunity cost of patient's time—appropriate treatment | 3.20 | 4.47 | 2.89–4.74 | Gamma (0.51, 0.07) | [41] |
| cCLTapp | Opportunity cost of carer's time—appropriate treatment | 1.44 | 2.71 | 1.24–1.65 | Gamma (151.73, 0.009) | [41] |
| cOPEnot | Patient out of pocket expenditure—inappropriate treatment | 5.61 | 17.16 | 4.32–6.91 | Gamma (14.77, 0.38) | [41] |
| cPLTnot | Opportunity cost of patient's time—inappropriate treatment | 4.15 | 4.90 | 3.78–4.52 | Gamma (99.01, 0.04) | [41] |
| cCLTnot | Opportunity cost of carer's—inappropriate treatment | 1.85 | 2.42 | 1.67–2.03 | Gamma (80.67, 0.02) | [41] |
| cANTI | Cost per full course of antibiotics | 0.12 | 0.02 | 0.11–0.12 | Gamma (25, 0.005) | [41] |
| cAMR | Cost of AMR to society per full course of antibiotics | 4.39 | 0.88 | 2.67–6.11 | Gamma (25, 0.18) | [42] |
| **Probability Parameter** | **Parameter Description** | **Probability*** | **95% CI** | | **Distribution (α, β) +** | **Source** |
| pBac | Prevalence of bacterial infection among malaria-negative patients | 0.10 | 0.09–0.10 | | Beta (15, 142) | [14] |
| pDIAGanti | Probability of being prescribed antibiotic under current clinical practice | 0.56 | 0.56–0.56 | | Beta (2479, 1912) | [30] |
| pSENSpoccrp | Sensitivity of POCCRP test for detection of CRP concentration at 10 mg/L | 0.95 | 0.87–0.97 | | Beta (86, 5) | [22] |
| pSPECpoccrp | Specificity of POCCRP test for non-detection of CRP concentration of less than 10 mg/L | 0.98 | 0.92–1.00 | | Beta (75, 2) | [22] |
| pBACcrp>10 | Probability of bacterial infection patient having CRP concentration of 10 mg/L | 0.95 | 0.92–0.97 | | Beta (404, 21) | [21] |
| pOTHcrp<10 | Probability of non-bacterial infection patient having CRP concentration of less than 10 mg/L | 0.49 | 0.46–0.53 | | Beta (464, 483) | [21] |

* Value used for the deterministic model.

+ Distribution for PSA.

Alpha (α) and Beta (β) values for Gamma distributions calculated as below (where $\bar{x}$ denotes mean and $\sigma\bar{x}$ denotes standard error of the data):

$$\alpha = \left(\frac{\bar{x}}{\sigma_{\bar{x}}}\right)^2$$

$$\beta = \frac{\sigma_{\bar{x}}^2}{\bar{x}}$$

Alpha (α) and Beta (β) values for Beta distributions are assigned based on counts of events of interest as a proportion of total sample. Standard error ($\sigma\bar{x}$) is calculated as below:

$$\sigma_{\bar{x}} = \sqrt{\frac{\alpha\beta}{(\alpha+\beta)^2 + (\alpha+\beta+1)}}$$

assigned to probability parameters; alpha and beta values were derived from counts of events of interest as a proportion of total sample [35]. Gamma distributions were assigned to all cost parameters, accounting for the skewed nature of cost data bounded by 0, derived from the

mean and standard error of the sampling distribution [35]. The simulations for the two different cost perspectives (health care delivery and societal) and scenario analysis (societal including economic cost of AMR) are presented as cost-effectiveness planes [35]. Expected outcomes for the POCCRP intervention compared to the current clinical practice were expressed as monetary net benefits for a specified value of WTP with the option with the highest net benefit being considered most cost-effective. The model was run for values of WTP from $0 to $200 per correctly treated patient at intervals of $0.50 for all cost perspectives. The output for each trial setting was presented graphically as cost-effectiveness acceptability curves [35].

## Scenario analysis

A scenario analysis was undertaken to attempt to incorporate the economic cost of AMR per prescription of antibiotics. The scenario analysis builds on the societal perspective, incorporating an additional cost parameter (economic cost of AMR per prescription). The economic cost of AMR included the additional cost of treating patients with resistant infections and the indirect productivity losses due to excess mortality attributable to resistant infections [42]. The estimate for this parameter was for a broad-spectrum penicillin (BSP), assumed to drive resistance in 5 pathogens in Thailand. A BSP was chosen as the most likely antibiotic to be used as first line treatment in Afghanistan.

## Results

### Deterministic ICER

Table 2 illustrates the results of the deterministic analysis. From a healthcare delivery perspective, the base case deterministic ICER is $14.33 per additional correctly treated febrile malaria-negative patient. From a broader, societal perspective, the ICER falls to $11.22. A scenario analysis including the economic cost of AMR reduces the ICER further to $9.60.

Population for the study is 4,391 patients. Compared to current clinical practice, there was an incremental positive effect of 421 patients treated correctly (9.6% of population) with the POCCRP intervention. Of the population (4,391), 45% (1,966) patients received the correct treatment under current clinical practice; while under the POCCRP intervention, 54% (2,387) received correct treatment. The introduction of the POCCRP intervention resulted in a 6% overall reduction in antibiotic prescriptions (137 / 2,479) across the study population. More significantly, there was a 12% reduction (279 / 2,242) in incorrect antibiotic prescriptions when comparing the POCCRP intervention to current clinical practice.

The decision tree model with all costs and probabilities assigned can be seen in S3 Appendix: populated decision tree.

### Sensitivity analysis

**Univariate.**   Univariate sensitivity analysis shows there were 12 variables that had an effect greater than $0.50 on the deterministic ICER when varied between their upper and lower bounds. Tornado plot in Fig 3 illustrates the range effect of the 12 variables.

**Table 2. Deterministic ICERs for healthcare delivery, societal and scenario analysis.**

| Perspective | Current clinical practice total cost for population ($) | POCCRP total cost ($) | Incremental cost ($) | Current clinical practice correctly treated (Patients) | POCCRP correctly treated (Patients) | Incremental effect (Patients) | ICER ($) |
|---|---|---|---|---|---|---|---|
| Healthcare delivery | 1,653 | 7,689 | 6,036 | 1,966 | 2,387 | 421 | 14.33 |
| Societal | 46,487 | 51,212 | 4,725 | | | | 11.22 |
| Scenario analysis | 57,371 | 61,413 | 4,042 | | | | 9.60 |

**Univariate sensitivty tornado: scenario analysis (societal perspective, including cost of AMR)**

**Fig 3. Univariate sensitivity tornado diagram–scenario analysis: Societal perspective (including cost of AMR).**

The probability of being treated with an antibiotic in current clinical practice was the parameter that had the greatest impact on ICER (range $20.82). Cost of the POCCRP test kit was the parameter that has the second greatest impact on ICER (range $10.91). Specificity of bacterial infection being detected at 10mg/L (i.e. the probability that non-bacterial infection does not have CRP concentration of 10 mg/L) followed with a range of effect of $10.71. The specificity and sensitivity of the POCCRP test kit followed as the fourth and fifth in terms of range effect on ICER (respective ranges: $8.06 and $7.19).

Table 3 shows the lower and upper bounds used to vary parameters for the univariate sensitivity analysis.

**Probabilistic sensitivity analysis.** Of the population (4,391), the probabilistic model indicated that 1,966 (45%) patients received the correct treatment under current clinical practice; while with the POCCRP intervention, the model indicated that 2,396 (55%) received correct treatment. Fig 4 is the cost-effectiveness for results from the PSA for the healthcare delivery

**Table 3. Univariate sensitivity analysis–scenario analysis: Societal perspective (including cost of AMR).**

| Parameter | Deterministic model | Parameter Lower Bound (LB) | Parameter Upper Bound (UB) | Basis for bounds | ICER parameter LB | ICER parameter UB | Range |
|---|---|---|---|---|---|---|---|
| Probability of being prescribed antibiotic under current clinical practice | 0.56 | 0.51 | 0.61 | + / - 5% | $25.31 | $4.49 | $20.82 |
| Cost of the POCCRP test kit | $1.00 | $0.61 | $1.39 | 95% CI | $4.14 | $15.05 | $10.91 |
| Specificity of non-bacterial infection patient having CRP concentration of less than 10 mg/L | 0.49 | 0.46 | 0.53 | 95% CI | $15.80 | $5.09 | $10.71 |
| Specificity of POCCRP test for non-detection of CRP concentration of less than 10 mg/L | 0.98 | 0.92 | 1 | 95% CI | $16.20 | $8.13 | $8.06 |
| Sensitivity of POCCRP test for detection of CRP concentration at 10 mg/L | 0.95 | 0.87 | 0.98 | 95% CI | $4.97 | $12.16 | $7.19 |
| Patient out of pocket expenditure—inappropriate treatment | $5.61 | $4.32 | $6.91 | 95% CI | $10.89 | $8.57 | $2.32 |
| Patient out of pocket expenditure—appropriate treatment | $3.82 | $2.89 | $4.74 | 95% CI | $8.67 | $9.88 | $1.21 |
| Cost of AMR to society per full course of antibiotics | $4.39 | $2.67 | $6.11 | 95% CI | $10.16 | $9.04 | $1.12 |
| Prevalence of bacterial infection among malaria negative patients | 0.10 | 0.05 | 0.15 | + / - 5% | $9.90 | $8.80 | $1.10 |
| Overhead cost of running the pharmacy/dispensary at the healthcare facilities | 1.33 | 1.29 | 1.38 | 95% CI | $9.13 | $10.06 | $0.93 |
| Opportunity cost of patient's time—appropriate treatment | $3.08 | $2.70 | $3.47 | 95% CI | $9.21 | $9.98 | $0.77 |
| Opportunity cost of patient's time—inappropriate treatment | $4.15 | $3.78 | $4.52 | 95% CI | $9.97 | $9.23 | $0.74 |
| Opportunity cost of carer's time—appropriate treatment | $1.44 | $1.24 | $1.65 | 95% CI | $9.39 | $9.80 | $0.41 |
| Sensitivity of bacterial infection patient having CRP concentration of 10 mg/L | 0.95 | 0.92 | 0.97 | 95% CI | $9.84 | $9.44 | $0.39 |
| Training to administer and interpret the test | $0.03 | $0.02 | $0.04 | 95% CI | $9.43 | $9.76 | $0.33 |
| Opportunity cost of carer's time—inappropriate treatment | $1.85 | $1.67 | $2.03 | 95% CI | $9.13 | $9.41 | $0.28 |
| Cost per full course of antibiotics in Afghanistan | $0.12 | $0.11 | $0.12 | 95% CI | $9.60 | $9.60 | $0.00 |
| Outpatient services delivery cost | $0.28 | $0.25 | $0.31 | 95% CI | $9.60 | $9.60 | $0.00 |

perspective. The diagram shows the uncertainty in incremental effect on the x-axis and incremental cost on the y-axis. The mean incremental effect is 9.8%, showing that the intervention is likely to be more effective than current clinical practice. On average, the POCCRP intervention is more expensive with an incremental cost of $1.43.

For societal perspective, the mean incremental effect was unaffected as the societal cost does not impact upon the measure of outcome. For both the societal perspective and the scenario analysis including the economic cost of AMR, the POCCRP intervention was on average more expensive. In the societal perspective, average incremental cost was $1.13; when including cost of AMR in the scenario analysis, average incremental cost fell to $0.96. Figs 5 and 6 are cost-effectiveness planes with results from the PSA of the societal cost perspective and scenario analysis including economic cost of AMR. There was greater uncertainty with regard cost, compared to the healthcare delivery cost perspective. Data points were both positive and negative in terms of cost, thus in some simulations the intervention was dominant over current clinical practice (i.e. it is more effective and less costly). For the societal cost perspective 6% of simulations (60/1000) estimated POCCRP intervention to be less costly compared to the

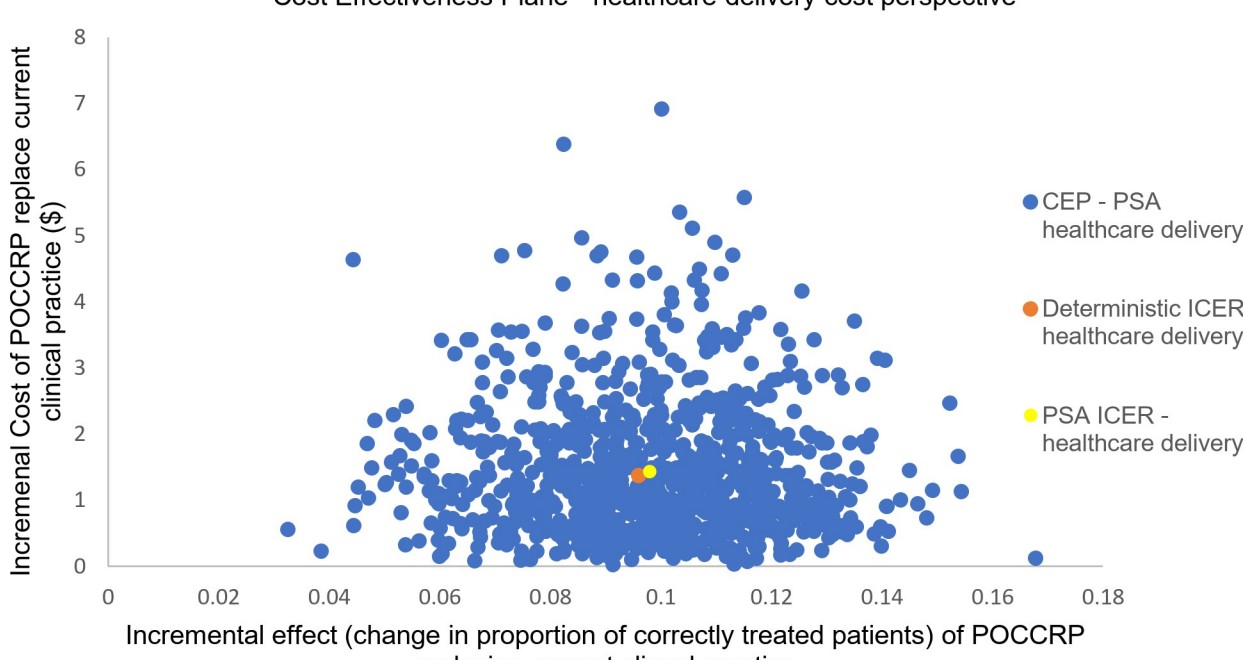

**Fig 4. Probabilistic sensitivity analysis (health sector perspective): Scatter plot of incremental health sector cost and incremental effect resulting from replacing current clinical practice by POCCRP testing.**

current clinical practice. For the scenario analysis (societal perspective including economic cost of AMR), this rose to 13% of simulations (130/1000) estimating POCCRP intervention to be less costly compared to current clinical practice.

The uncertainty in the cost-effectiveness planes was summarised in the cost-effectiveness acceptability curves (Fig 7), showing the probability that introduction of the POCCRP intervention is cost-effective according to different levels of WTP for an appropriately treated patient for each of the cost perspectives. For healthcare delivery perspective, if WTP for additional correctly treated patient is zero, there was a 0% probability that the intervention can be considered cost effective. However, for societal perspective and the scenario analysis including cost of AMR, the model shows 6% and 13% respective probabilities that the intervention would be considered good value at WTP of zero. For healthcare delivery perspective, if we increase WTP to $12.50, the probability of cost effectiveness increased to 50%. For the societal cost perspectives, probability of cost effectiveness remains higher at 61% and 66% respectively. For healthcare delivery cost, WTP of $30.50 showed a 90% probability that the intervention was cost effective. For societal cost perspective and the scenario analysis including cost of AMR, 90% probability of cost effectiveness was not reached until WTP of $27.50 and $26.00 respectively.

## Discussion

The cost-effectiveness analysis of POCCRP test for malaria-negative febrile patients shows that replacing current clinical practice by POCCRP intervention is likely to be more effective compared to current clinical practice, but also likely to be more costly. Compared to current clinical practice, the introduction of POCCRP testing is likely to be associated with a reduction of incorrect antibiotic prescriptions among malaria-negative febrile patients.

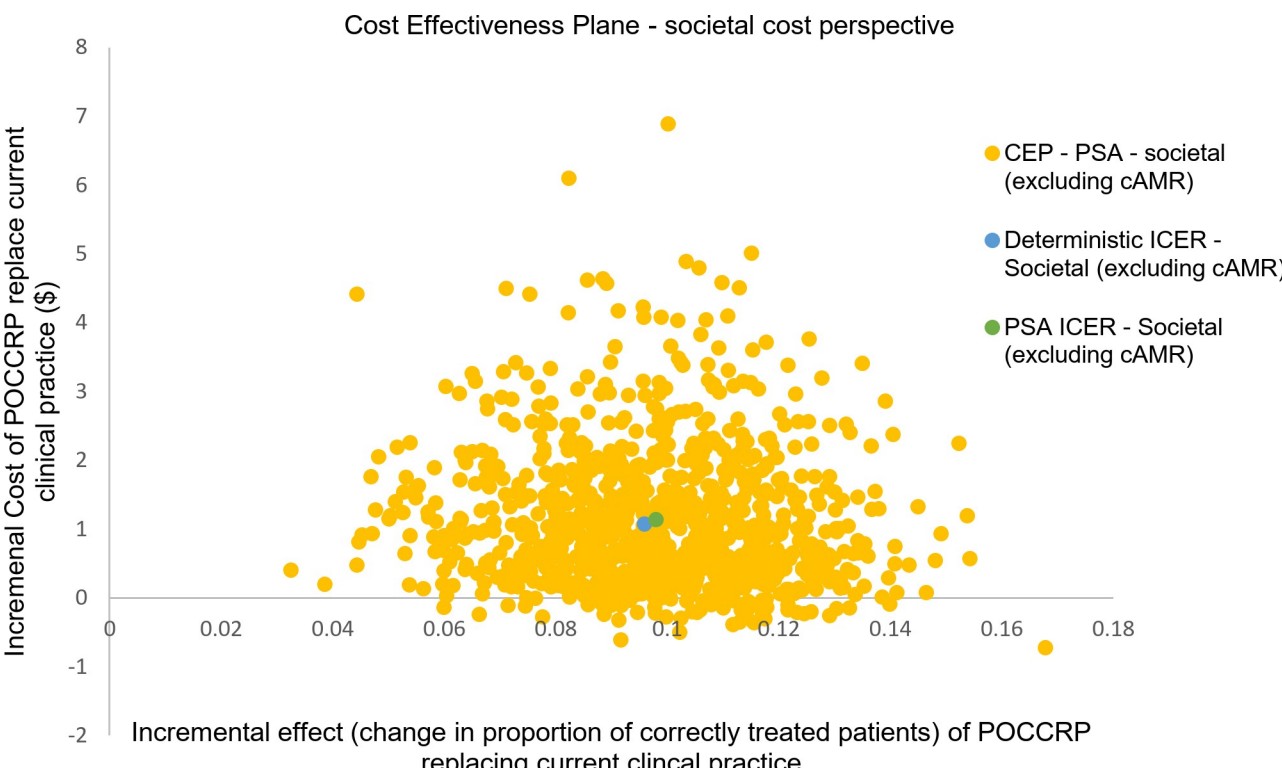

**Fig 5. Probabilistic sensitivity analysis (societal perspective): Scatter plot of incremental health sector cost and incremental effect resulting from replacing current clinical practice by POCCRP testing.**

Health service willingness to pay data is not available for correct treatment of malaria-negative febrile patients. The only comparable study that estimates health service willingness to pay (WTP) was conducted in the USA, with a threshold of $43 per antibiotic prescription safely avoided [46]. While the measure of outcome for the study was different, a reduction in unnecessary antibiotic prescriptions of 279 was observed at incremental costs for the intervention of $6,036 (healthcare perspective), $4,725 (societal perspective), and $4,042 (scenario analysis). This equates to a cost per unnecessary antibiotic prescription avoided of $21.63, $16.93 and $14.49, respectively. While this comparison is from a high-income country and is only from one study, it provides positive indication the POCCRP intervention could be cost-effective and that further research is warranted. WTP for interventions to reduce unnecessary antibiotic use are further complicated because when considering WTP, there are no immediate health benefits to the patient–if an antibiotic is given unnecessarily, it will have little effect on treatment outcomes compared to no antibiotic given in the first place. Thus, a patient would only likely be willing to pay for the intervention if it cost less than the treatment. So, policy makers (donors and governments) must make a decision to incur costs now to off-set *future* health impacts. Given the global health security threat of AMR, it is possible that bilateral or multilateral donors would consider funding interventions to reduce unnecessary use of antibiotics. Research is required into WTP for correctly treated malaria-negative febrile patients to be able to say with any certainty that the intervention would be cost effective.

There is debate as to what CRP concentration should be used to determine if a bacterial infection is likely and an antibiotic required [18, 21, 47]. 10 mg/L is a relatively low CRP concentration threshold to consider prescription of an antibiotic as compared to what is used in

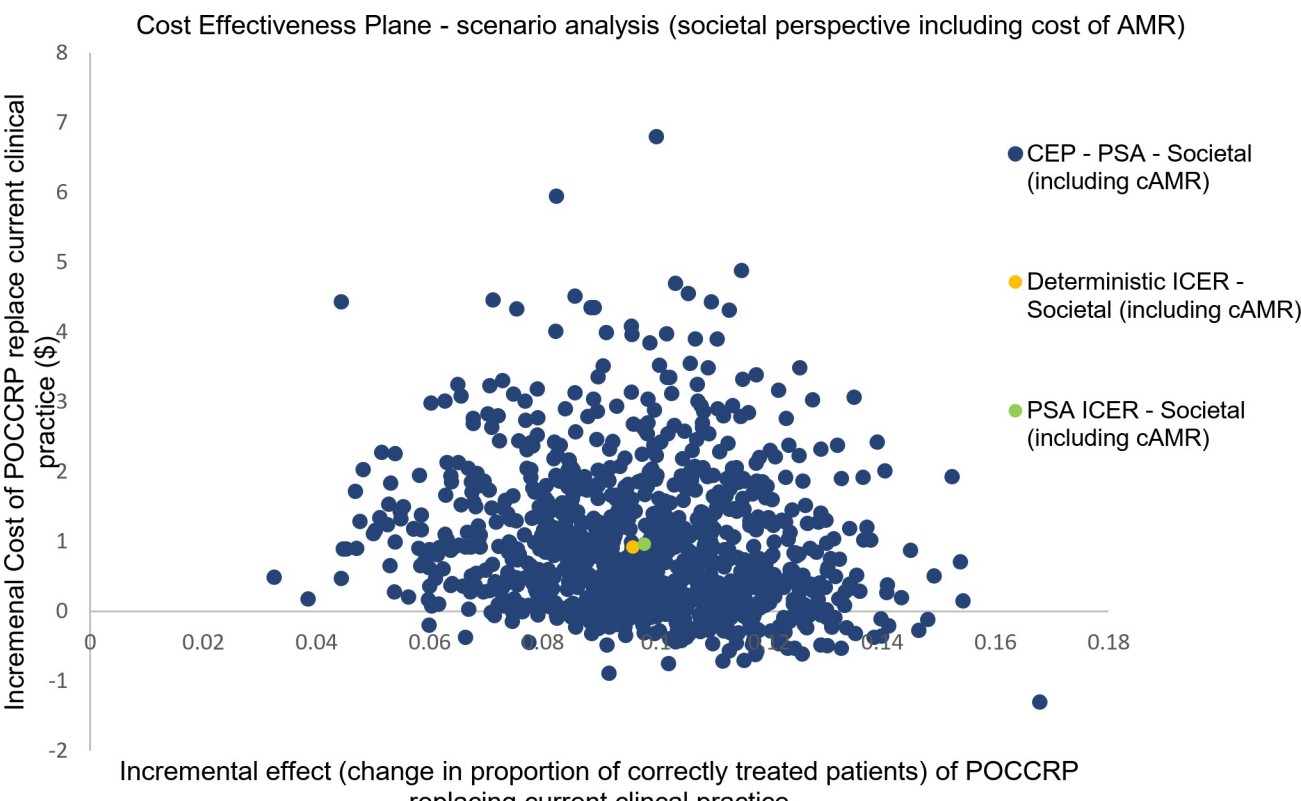

Cost Effectiveness Plane - scenario analysis (societal perspective including cost of AMR)

**Fig 6. Probabilistic sensitivity analysis (scenario analysis—societal perspective including economic cost of AMR): Scatter plot of incremental health sector cost and incremental effect resulting from replacing current clinical practice by POCCRP testing.**

high-income countries. For example in the United Kingdom, National Institute of Health and CARE Excellence (NICE) advises a threshold of 20mg/L [47]. A lower threshold means there is less likelihood of missing bacterial infections and in settings where there is little recourse for emergency care this could be more appropriate. However, it also means that many more non-bacterial infections are incorrectly diagnosed as bacterial infections. At a 20 mg/L threshold, the sensitivity of detecting bacterial infections decreases to 86% compared to 95% at 10 mg/L, and the specificity increases to 67% compared to 49% at 10 mg/L [21]. CRP concentration threshold to consider prescription of an antibiotic would be an important factor for introduction of POCCRP and different thresholds could be tested alongside each other in any future studies.

There are POCCRP tests that allow for interpretation of CRP concentration, indicating bands of strength of concentration, such as the WD-23 and bioNexia CRPplus [26]. These could be advantageous as they would allow a more nuanced interpretation by the healthcare worker of the test, which they could then combine with patient symptoms to determine if an antibiotic is required. Further research should consider different POCCRP tests, looking at differences in functionality and accuracy, but also taking into account that increased complexity to interpret results could adversely affect overall effectiveness.

Economic cost of AMR per prescription of antibiotics is an important element to any study looking at cost-effectiveness of interventions to reduce unnecessary use of antibiotics. In the model, we see ICER reduces by $1.62 between societal cost perspective ($11.22) and the scenario analysis where economic cost of AMR is included ($9.60). The economic cost of AMR

**Cost Effectiveness Acceptability Curves - healthcare perspective**

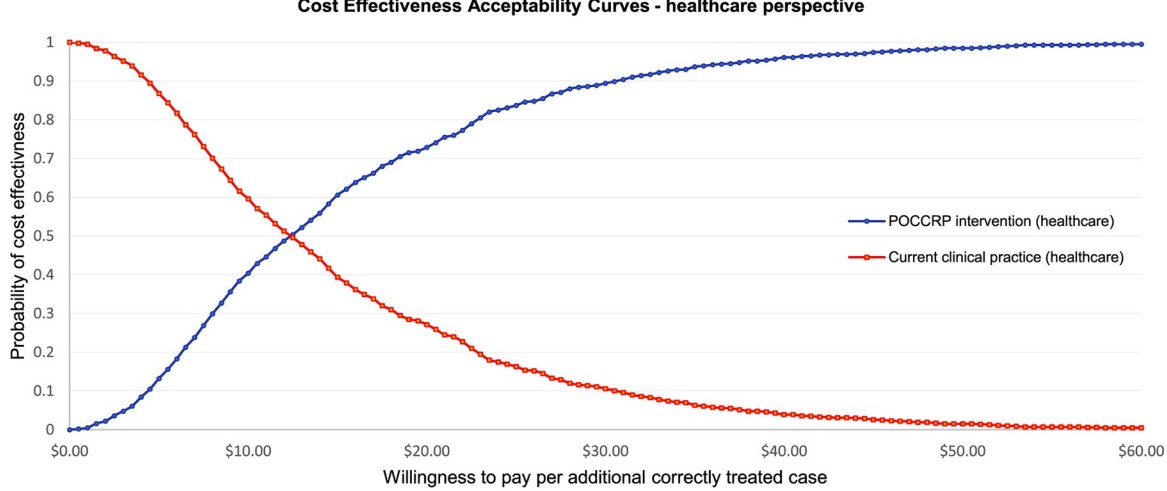

**Cost Effectiveness Acceptability Curves - societal perspective**

**Cost Effectiveness Acceptability Curves - scenario analysis (societal perspective + cost of AMR)**

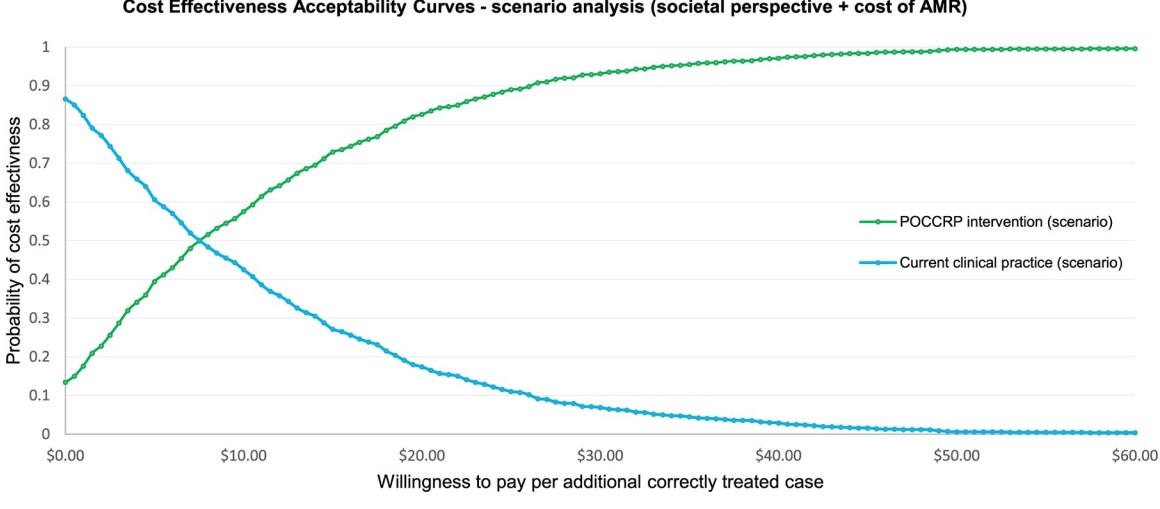

**Fig 7. Cost-effectiveness acceptability curves.**

data used in the model is taken from one study based in Thailand [42] as there is limited research that estimates cost of AMR by antibiotic prescription. The estimate used for this parameter in the scenario analysis should be considered conservative as the source study is "narrowly defined as the incremental cost of treating patients with resistant infections as compared with sensitive ones, and the indirect productivity losses due to excess mortality attributable to resistant infections" and the source considers cost of AMR for five pathogens only [42]. Further research is required to understand economic costs of AMR resulting from a prescription of antibiotics so that this significant cost can be considered for interventions seeking to reduce unnecessary use of antibiotics.

A further limitation of our modelling is that the time horizon is short (28 days). Whilst this is sufficient to capture the endpoint of interest for the CEA, we acknowledge that it does not sufficiently capture all the long-term benefits of reduced AMR, where quality of life gains and mortality reduction may be realised. To achieve a robust estimate of the longer-term cost-effectiveness would require a highly complex decision analysis model, over a lifetime horizon, potentially restructured as a cost-utility analysis reporting QALY gains or DALYs averted. Whilst it was not possible to achieve this within the scope of the current work, this is a promising area for future research.

While the study indicates that POCCRP intervention would likely be more effective compared to current clinical practice in Afghanistan, there is limited scope to generalise the results to other settings. The reason for this is that there is substantial heterogeneity in clinical practice, health system organisation, and unit costs of treatments across different settings. Further, differing local epidemiological factors would impact upon cost-effectiveness of the intervention (e.g. prevalence of bacterial infections in non-malaria febrile patients). Given positive indications of the effectiveness of the intervention, further research is warranted in other malaria endemic countries where POCCRP could contribute to reducing inappropriate prescriptions of antibiotics.

The CEA has some important limitations. Household cost parameters accounted for a significant amount of cost (84% in the societal cost perspective). Household data used included malaria-positive and malaria-negative patients from the malaria RDT CEA study [41], as cost data between the two could not be disaggregated from the primary dataset. Further, the dataset had high variation in costs throughout the sample (observed by relatively high standard deviations for these parameters). The model could be strengthened by more specific research into patient and carer costs for malaria-negative patients only. Aetiology of non-malarial fever is an area where there is limited data [15, 37–40] but this parameter is very important to understand the effectiveness of POCCRP to reduce unnecessary antibiotic prescriptions. For the study, data from Afghanistan was provided by researchers of an unpublished study [14]. The model assumed that POCCRP test results will be adhered to in all cases by both healthcare worker and patient but evidence from use of introduction of RDTs suggests this is unrealistic, with non-adherence to negative tests being as high as 61% [27]. Future research into POCCRP test cost-effectiveness should measure and account for non-adherence to negative test results.

This study builds on initial work in the area of POCCRP and is the first CEA to review POCCRP to reduce inappropriate use of antibiotics in an LMIC. The study benefits from using primary data from costs (healthcare and societal) for introduction of malaria RDTs in Afghanistan.

## Conclusion

The introduction of POCCRP tests could be effective to increase number of correctly treated malaria-negative patients in Afghanistan and reduce unnecessary use of antibiotics compared

to current clinical practice. Whether it can ultimately be considered good value for money is dependent on WTP of policy makers and other possible funders. This study is based on a combination of trial data for introduction of malaria RDTs and secondary sources on POCCRP accuracy. To establish whether there is a place for POCCRP testing in clinical guidelines in LMICs, primary data in a trial setting would be required. A trial would need to assess clinical, microbiological and economic outcomes, as well as WTP of policy makers and donors. The results of the study suggest that further research is warranted into the cost effectiveness of POCCRP tests in malaria-negative patients to avoid unnecessary prescription of antibiotics in Afghanistan. Research could also be expanded to other malaria endemic countries.

## Supporting information

**S1 Appendix. Working file of model and sensitivity analysis.**
(XLSM)

**S2 Appendix. Model parameters and sources.**
(DOCX)

**S3 Appendix. Populated decision tree.**
(TIFF)

## Acknowledgments

The paper was initially developed as part of an MSc dissertation by the lead author at the University of Aberdeen. The authors acknowledge the inputs from researchers into the primary data collection in 2009–2012 and CEA study for the introduction of Malaria RDTs in Afghanistan; not all of these authors met criterion for authorship on this paper.

## Author Contributions

**Conceptualization:** Simon Dickinson, Huey Yi Chong, Toby Leslie, Mark Rowland, Kristian Schultz Hansen, Dwayne Boyers.

**Formal analysis:** Simon Dickinson.

**Investigation:** Toby Leslie, Mark Rowland, Kristian Schultz Hansen.

**Methodology:** Simon Dickinson, Huey Yi Chong, Dwayne Boyers.

**Writing – original draft:** Simon Dickinson.

**Writing – review & editing:** Simon Dickinson, Huey Yi Chong, Toby Leslie, Mark Rowland, Kristian Schultz Hansen, Dwayne Boyers.

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
