## [Decision Letter · Decision Letter 0]

23 Jul 2021

PONE-D-21-01807

Cost-effectiveness of point-of-care C-Reactive Protein test compared to current clinical practice as an intervention to improve antibiotic prescription in malaria-negative patients in Afghanistan.

PLOS ONE

Dear Dr. Dickinson,

Thank you for submitting your manuscript to PLOS ONE. After careful consideration, we feel that it has merit but does not fully meet PLOS ONE’s publication criteria as it currently stands. Therefore, we invite you to submit a revised version of the manuscript that addresses the points raised during the review process.

We look forward to receiving your revised manuscript.

Kind regards,

Ismaeel Yunusa, PharmD, PhD

Academic Editor

PLOS ONE

Journal Requirements:

"The authors received no specific funding for this article.

The primary data collection (2009 - 2012) for the malaria RDT IRT (30) and CEA (41)

in Afghanistan was funded by the Bill and Melinda Gates Foundation through a grant to

the London School of Hygiene and Tropical Medicine. Data on aetiological agents of

disease (14) was collected with funding from the UK Defence Science and Technology

Laboratory."

We note that one or more of the authors are employed by a commercial company: "Mott MacDonald Ltd;"

Additional Editor Comments (if provided):

The reviewers made essential comments that need to be addressed before we reconsider this manuscript. For your cost-effectiveness Acceptability Curve (CEAC), please show both interventions within the curve. It will be more informative, even when it takes having three graphs with panels A, B, and C for the perspectives you are interested in presenting. Alternatively, present a Cost-effectiveness Acceptability Frontier (CEAF), which would show only the optimal strategy for each perspective within the same graph.

Also, report your revised manuscript using the CHEERS guideline and submit a CHEERS checklist along with the updated manuscript.

Reviewers' comments:

Reviewer's Responses to Questions

**Comments to the Author**

1. Is the manuscript technically sound, and do the data support the conclusions?

Reviewer #1: Yes

Reviewer #2: Partly

Reviewer #3: Yes

2. Has the statistical analysis been performed appropriately and rigorously? 

Reviewer #1: Yes

Reviewer #2: Yes

Reviewer #3: Yes

3. Have the authors made all data underlying the findings in their manuscript fully available?

Reviewer #1: Yes

Reviewer #2: Yes

Reviewer #3: Yes

4. Is the manuscript presented in an intelligible fashion and written in standard English?

Reviewer #1: Yes

Reviewer #2: Yes

Reviewer #3: No

5. Review Comments to the Author

Reviewer #1: In this study, a decision analytic model was used to examine the cost-effectiveness of POCCRP to improve antibiotic prescriptions in malaria-negative patients in Afghanistan. The study concluded that POCCRP tests could improve antibiotic prescribing in Afghanistan. Overall, this economic evaluation has been well conducted and I have a few comments.

The cost-effectiveness of POCCRP testing has been studied in many settings previously. However, not much in a LMIC setting as a result, this study adds a different dimension to the literature on the cost-effectiveness of POCCRP testing.

The perspective of the study should be made explicit in the abstract. Presently, this is not really clear in the abstract.

With respect to the outcome measure that has been used, my main concern was the fact that there is no threshold to determine whether the intervention is cost-effective or not. It would be interesting to consider other outcome measures as well e.g. the QALY or the DALY.

Great to include the scenario analysis which accounts for the cost of antibiotic resistance. However, it is not really clear how this cost was included in the model.

More information on the generalisability of the findings to other settings is needed

There is an error on line 142 page 14 i.e. last sentence under the heading “POCCRP test intervention and diagnostic accuracy” which needs to be deleted. In addition, page numbers need to be sorted out

Figure 2 is repeated line 161 on page 15.

Line 192 is not clear

Reviewer #2: This is a well written paper evaluating the cost-effectiveness of a point of care C-Reactive Protein (POCCRP) test for bacterial infection in Afghanistan. A decision model was developed, with a result of a 12% reduction in inappropriate antibiotic prescriptions for POCCRP compared to standard practice, hence potentially reducing the risk of antibiotic microbial resistance in the future, with a cost per additional correctly treated case of $14.33 USD from a health care perspective.

I have a number of comments:

1) In the abstract the population size would be helpful to put the results in context.

2) In regards to (1) I wasn't sure why the population size of 4391 had been chosen. It was the number of people who happened to be in the trial, but I wasn't sure if this was a useful number for a decision maker. Would this represent the size of an average region? Using a population size of 1,000 though, for example, would make it easier for decision makers to multiple up to their specific region.

3) Overall, I wasn't clear how the data from the trial were used to inform the model other than to be the population selected, given, from what I understood from reading the paper, the trial did not involve POCCRP? Why and how the trial was used could be made clearer.

4) On page 14 of the methods, I found the reference standard paragraph a little confusing. I understand diagnostic tests to be tests that are used when making diagnostic decisions regarding a disease. My understanding was the POCCRP are not diagnostic tests but clinical decision making tools to help clinicians decide on the potential best course of treatment (as opposed to screening tools that help identify an increased risk of a disease and hence the need to then be escalated to a diagnostic test).

5) In relation to (4) above, using lab tests as a reference standard for 10mg/L makes sense to me. Using the NycoCard Reader II as a reference does not. To me the NycoCard Reader II sounds like a POCCRP assay device that is used for quality benchmarking. Is this correct? Quality benchmarking is not the same as the actual performance of the device being used, or the same as a reference standard. I might be misreading this whole section, but overall it would help to make clear what device is being used and what performance it has been CE marked at.

6) Table 1: The prevalence of bacterial infection among malaria-negative patients appears to be treated as a probability and not a rate as it should be. If the rate has been converted to a probability this should be made explicit.

7) My biggest concern with the model was that I wasn't clear how the negative impact of missing bacterial cases and not prescribing antibiotics had been incorporated into the model. Although details are given on correct prescribing in the results, no details are provided on the number of bacterial cases missed and no discussion of the potential negative health impact of missing those cases.

8) I wasn't clear why the deterministic analysis was reported as the base case as my understanding is that best practice is to report the probabilistic analysis as the base case given that it directly relates to the assessment of uncertainty.

9) Results page 17 - it would help to have the total population reported right as the start of the results section, particularly given the strange number chosen.

10) Discussion page 23 - "a reduction in unnecessary antibiotic prescriptions of 279 was observed". I wasn't sure what this meant sorry? Is 279 good? 279 per what? There is also no unit given for the denominator in the sentence. Incremental cost per what?

11) Overall, I wasn't convinced by the argument used for the willingness to pay threshold from the US, given that the the US and Afghanistan are polar opposites in many factors.

12) There was only limited discussion provided regarding if the assumptions made in the model are realistic for if the test was implemented into standard practice. My understanding with POCCRP tests is that a lot of work is required to implement these tests, ensure clinicians use them and to ensure that patients find them acceptable. I didn't feel this was adequately addressed in the paper. There was also no discussion of what model would be used to pay for the assay device - fixed upfront or monthly payments? How was this incorporated into the model? What impact would the cost of the device have on uptake given the limited finances in Afghanistan?

Reviewer #3: Summary: The grammatical mistakes detract from the description and message, but overall the methods part of this manuscript seems ok. Modelers seem to get more leeway when it comes to their distributional choices and how they account for uncertainty than empirical analyses. I am a little uneasy that so many of the parameter distributions come from a single study. I imagine it's likely that there is not much data from Afghanistan, though it might be nice to at least discuss how these values compare with other places and ensure that those values are included in distributions.

1. There are some grammar and editing mistakes sprinkled throughout this manuscript. At times this is not very serious, but other times it makes the article challenging to understand, e.g. line 170. Manuscripts at PLOS ONE can be rejected based on criterion #5 (The article is presented in an intelligible fashion and is written in standard English). The authors need to read this manuscript more closely.

2. (Figure 2) I don't understand the tree diagram. Why is the node for bacterial infection or other cause a circle? The definition in the note for Figure 2 says circle nodes are chance nodes, but I would expect whether or not the person has an infection to be nonrandom.

3. (line 252) One thousand simulations is pretty low. It's questionable whether that allows you to find some of the more rare situations and computing power is pretty easy to come by. I encourage using 10,000 simulations.

4. Was anything done in the simulations to ensure that the probabilities would sum to 1?

5. (line 254) I believe you mean "binary" instead of "binomial".

6. (line 262) I didn't understand the WTP range that was chosen for this. In the introduction, the WTP threshold is much higher, though that's per QALY.

7. (Figure 3) Why are the values centered around $10? I'm guessing these are the outcome values of the simulations and the range of them.

6. PLOS authors have the option to publish the peer review history of their article (what does this mean?). If published, this will include your full peer review and any attached files.

Reviewer #1: No

Reviewer #2: **Yes: **Rachael Hunter

Reviewer #3: No

---

## [Author Response · Author response to Decision Letter 0]

14 Sep 2021

We thank the reviewers and the academic editor for their comments, which have undoubtedly improved the quality of the manuscript. We provide specific response to all reviewer comments in the "Response to Reviewers" letter which is included as part of this re-submission.

---

## [Editor Report · Decision Letter 1]

24 Sep 2021

Cost-effectiveness of point-of-care C-Reactive Protein test compared to current clinical practice as an intervention to improve antibiotic prescription in malaria-negative patients in Afghanistan.

PONE-D-21-01807R1

Dear Dr. Dickinson,

We’re pleased to inform you that your manuscript has been judged scientifically suitable for publication and will be formally accepted for publication once it meets all outstanding technical requirements.

Kind regards,

Ismaeel Yunusa, PharmD, PhD

Academic Editor

PLOS ONE
---

## [Editor Report · Acceptance letter]

27 Oct 2021

PONE-D-21-01807R1 

Cost-effectiveness of point-of-care C-Reactive Protein test compared to current clinical practice as an intervention to improve antibiotic prescription in malaria-negative patients in Afghanistan. 

Dear Dr. Dickinson:

I'm pleased to inform you that your manuscript has been deemed suitable for publication in PLOS ONE. Congratulations! Your manuscript is now with our production department. 

Kind regards, 

on behalf of

Dr. Ismaeel Yunusa 

Academic Editor

PLOS ONE